# Solvent-Free Mechanochemical Synthesis of ZnO Nanoparticles by High-Energy Ball Milling of ε-Zn(OH)_2_ Crystals

**DOI:** 10.3390/nano11010238

**Published:** 2021-01-18

**Authors:** Gil Otis, Michal Ejgenberg, Yitzhak Mastai

**Affiliations:** Department of Chemistry and the Institute of Nanotechnology, Bar-Ilan University, Ramat-Gan 52900, Israel; gotis89@gmail.com (G.O.); michaldomb@gmail.com (M.E.)

**Keywords:** zinc oxide, ball milling, inorganic chemistry, nanomaterials

## Abstract

A detailed investigation is presented for the solvent-free mechanochemical synthesis of zinc oxide nanoparticles from ε-Zn(OH)_2_ crystals by high-energy ball milling. Only a few works have ever explored the dry synthetic route from ε-Zn(OH)_2_ to ZnO. The milling process of ε-Zn(OH)_2_ was done in ambient conditions with a 1:100 powder/ball mass ratio, and it produced uniform ZnO nanoparticles with sizes of 10–30 nm, based on the milling duration. The process was carefully monitored and the effect of the milling duration on the powder composition, nanoparticle size and strain, optical properties, aggregate size, and material activity was examined using XRD, TEM, DLS, UV-Vis, and FTIR. The mechanism for the transformation of ε-Zn(OH)_2_ to ZnO was studied by TGA and XPS analysis. The study gave proof for a reaction mechanism starting with a phase transition of crystalline ε-Zn(OH)_2_ to amorphous Zn(OH)_2_, followed by decomposition to ZnO and water. To the best of our knowledge, this mechanochemical approach for synthesizing ZnO from ε-Zn(OH)_2_ is completely novel. ε-Zn(OH)_2_ crystals are very easy to obtain, and the milling process is done in ambient conditions; therefore, this work provides a simple, cheap, and solvent-free way to produce ZnO nanoparticles in dry conditions. We believe that this study could help to shed some light on the solvent-free transition from ε-Zn(OH)_2_ to ZnO and that it could offer a new synthetic route for synthesizing ZnO nanoparticles.

## 1. Introduction

For a long time, ZnO nanostructures have been attracting huge interest due to the prominent applications these materials have to offer. For example, ZnO is an important semiconductor with a relatively wide direct band gap (e.g., ∼3.3 eV at 300 K) [1], which makes it perfect for solar cells [2], photocatalysis [3,4], optical sensors, and other applications [5,6]. Moreover, its transparency in visible light led to the development of n-doped Al–ZnO coatings that are nowadays used as a cheap alternative to commercially available indium–tin–oxide transparent conductors [7,8]. ZnO nanoparticles are also used as UV-blockers in sunscreen [9], visible light photocatalysts [10], and antibacterial materials, as they generally release reactive oxygen species upon exposure to UV light in an aqueous solution [11,12].

Therefore, the synthesis of ZnO nano-powders is very desirable. Many synthetic methods are used to prepare these materials, including hydrothermal treatment [13,14,15], chemical bath deposition (CBD) [16,17], sol–gel-assisted synthesis [18,19], and others [20,21], most of which require solvation of ε-Zn(OH)_2_ crystals in an alkaline media. The accepted mechanism for the transformation of ε-Zn(OH)_2_ to ZnO is the solvation–precipitation mechanism, starting with the generation of solvated zincate ions and continuing with the precipitation of ZnO [22,23,24]: ε-Zn(OH)_2_ (s) + 2OH^−^ ⟶ [Zn(OH)_4_]^2−^ (aq) ⟶ ZnO (s) + H_2_O + 2OH^−^

Recently, an additional mechanism for the synthesis of ZnO was suggested [25]—a solid-state phase transformation:ε-Zn(OH)_2_ (s) ⟶ ZnO (s) + H_2_O

It was shown that even without the solvation of ε-Zn(OH)_2_, the phase transformation is possible by heating to relatively low temperature [25], and that with solvation of crystals, the transformation can occur through the solid pathway in parallel to the solution mechanism [26,27].

High-energy ball milling is a technique in which mechanical energy is used to grind down different powders; the grinding action happens during repeated collisions of the balls with the powder due to the rapid rotation of the mill [28]. In most cases, ball milling is used to break down bulk powders into fine powders with a smaller and more uniform size; however, mechanochemical processes can also occur during the milling, whereby mechanical energy is used to activate chemical reactions [28,29,30,31]. There are many examples where ball milling was used to execute various chemical reactions [32,33,34]. This solvent-free synthetic method could, therefore, be used to study the solid-state phase transformation of ε-Zn(OH)_2_ into ZnO nanoparticles.

In the study of ZnO, the ball milling technique was mainly used either to grind large micron-sized crystals into nanocrystalline material [35,36] or to grind Zn powder into ZnO [37,38,39]; however, mechanochemical synthesis of ZnO nanoparticles directly from ε-Zn(OH)_2_ appears to be novel. Here, we present such a synthesis of ZnO nanocrystals from ε-Zn(OH)_2_ micron-sized crystals; a simple ball milling process was used at ambient conditions. The milling was performed at very low speeds for short periods of time, and samples were collected from several points in the milling process in order to thoroughly investigate the transition step by step. The milling process was studied by looking at all the parameters of the materials: chemical structure, optical properties, average crystallite size, average nanocrystal aggregate size, and photocatalytic activity of the powders. 

## 2. Materials and Methods

### 2.1. Materials 

Sodium hydroxide pellets were purchased from Frutarom Ltd. (Haifa, Israel), zinc nitrate hexahydrate and methyl orange (MO) were acquired from Sigma Aldrich Co. (St. Louis, MO, USA), and ethanol (absolute) was obtained from Bio-Lab Ltd. (Jerusalem, Israel). 

### 2.2. Methods 

ε-Zn(OH)_2_ micron-sized crystals were prepared by titrating a super-concentrated (2.4 M) sodium hydroxide aqueous solution into a zinc nitrate aqueous solution (0.33 M) in an ice bath. The obtained product was washed several times with double distilled water and ethanol by centrifugation and dried using a vacuum pump to yield homogeneous white polycrystalline powder with an average crystal size of 2.42 ± 0.2 (Appendix A). The ε-Zn(OH)_2_ powder was taken to ball milling; generally, 100 mg of powder were placed in an 80 mL ZrO_2_ cup with steel casting, 1 mm diameter ZrO_2_ grinding balls were placed in the cup with a powder/ball mass ratio of 1:100, and 5 mL of ethanol was added to the cup and mixed in order to receive a homogeneous mesh of powder–ball mixture. The cups were sealed and placed into a Fritsch Pulverisette 7 planetary micro-mill, and milling cycles were performed, consisting of a 1 min run at 400 rpm, followed by a 10 min pause before the next run. In order to examine the milling process step by step, four samples were prepared and subjected to a different number of cycles—2, 5, 8, and 10 cycles. After milling, powders were washed with ethanol, dried, and characterized.

Crystallographic structures were measured by X-ray diffraction (XRD) measurements using a Bruker AXS D8 Advance diffractometer with Cu Ka (λ = 1.5418 Å) operating at 40 kV/40 mA and collecting from 2θ = 10° to 80°. Transmission electron microscopy (TEM) images were taken using a JEOL JEM-1400 microscope operated at 120 kV; samples were prepared by making a dilute suspension of the powders in isopropanol, placing them dropwise on a 400-mesh carbon-coated copper grid, and vacuum drying the grid. High-resolution TEM (HRTEM) images were taken using a W-200 kV JEOL JEM-2100 microscope operated at 200 kV. Thermogravimetric analysis (TGA) was performed with a Perkin–Elmer model Pyris 1 TGA instrument; samples were pre-heated from 30 to 100 °C at 40 °C/min, kept at 100 °C for 1 h to remove any humidity and excess ethanol adhered to the sample, and heated from 100 to 900 °C at 10 °C/min under N_2_ flow. X-ray photoelectron spectroscopy (XPS) analysis was carried out using a Nexsa spectrometer (England) equipped with a monochromatic micro-focused low power Al Ka X-ray source (photon energy 1486.6 eV). The binding energies of all elements were recalibrated by setting the CC/CH component of the C 1s peak at 285 eV. Measurements were carried out under an ultrahigh vacuum at a base pressure of 5 × 10^−10^ torr (no higher than 3 × 10^−9^ torr). 

Photocatalytic degradation of MO was studied with a 0.12 mM stock solution (2 mg/50 mL), which was prepared and kept in the dark. ZnO powders were suspended in MO stock solution (2 mg/mL) and stirred in the dark for 1 h to allow the formation of a stable ZnO–MO suspension and achieve better adhesion of the dye onto the nanocrystalline aggregates. After stirring, suspensions were taken to a Vilber Lourmat BLX254 UV crosslinker for the reaction, and a photocatalytic degradation assay was performed by exposing the samples to 254 nm UV light ($3\times 10^{-5}\;W/cm^{2}$) for 30 min. After the reaction, ZnO nanocrystals were filtered out, and the solutions were measured by UV-Vis spectroscopy using a V-570 Jasco UV spectrophotometer.

## 3. Results

### 3.1. Characterization 

#### 3.1.1. XRD Analysis

ZnO powders were prepared by subjecting ε-Zn(OH)_2_ crystals to 2, 5, 8, or 10 cycles of high-energy ball milling, and the diffraction patterns of the powders were examined by XRD, as described in detail above. Figure 1 presents the patterns obtained for the ε-Zn(OH)_2_ and milled powders. Surprisingly, even early on in the milling process, after only two cycles, we could identify the characteristic (1,0,0), (0,0,2), and (1,0,1) planes of zincite (AMCSD number [40]: 0005203) as well as the characteristic (1,1,0) and (1,0,1) planes of wülfinigite (ICSD number [41]: 50447). With increasing cycles, the ε-Zn(OH)_2_ planes disappeared and only those of ZnO remained (Figure 1A); the ZnO peaks became wider, indicating the presence of nanocrystallites. A more detailed graph of the dominant (1,0,1) plane of ZnO is presented in Figure 1B, demonstrating peak broadening that increases collinearly with increasing milling cycles.

ZnO nanocrystallite size was determined using XRD peak profiling analysis via the Scherrer and Williamson–Hall (WH) methods. The former, using the Scherrer equation, based on broadening of diffraction peaks, is the most popular and straightforward method for estimating average nanocrystalline size [42,43,44]. In order to correctly address the measured full width at half maximum (FWHM) values, the constant width of the instrument was subtracted by measuring the diffraction pattern of a standard LaB6 SRM 660c powder and creating a calibration curve for the FWHM with relation to the Bragg angle (Appendix A); as the LaB6 particle size and shape do not contribute to the peak width, the measured values arise from the instrument. The data were also de-convoluted using a Lorentzian function, and the corrected FWHM values and their corresponding Bragg angles were used to calculate the crystallite size according to the Scherrer method (Table 1) by averaging the six dominant and distinguishable ZnO planes—(1,0,0), (0,0,2), (1,0,1), (1,0,2), (1,1,0), and (1,0,3).

The WH method also contemplates the effect of strain on peak broadening and is considered a more accurate method for the estimation of crystallite size [45,46]. Assuming that the strain of the milled particles is uniform in all directions—in ball milling, stress is applied from all directions simultaneously and equally—the uniform deformation method (UDM) can be used to calculate the crystallite size and strain:(1)$$\beta \cos\left(\theta \right)=\;\left(\frac{K\lambda }{D}\right)+4\varepsilon \sin\left(\theta \right)$$
where  D is the crystallite size, K is the crystallite shape factor (usually taken as 0.9 for crystallites without a distinctive shape), λ is the X-ray wavelength (1.5418 Å), β is the FWHM of the diffraction peak in radians, ε is the lattice strain, and θ is the Bragg angle.

Hence, by plotting 4sin(θ) over βcos(θ) (Appendix A), the extrapolated intercept with the y-axis gives the crystallite size, and the slope of the curve is the strain (ε) [45], just like in the Scherrer method. For the calculation, we used the corrected FWHM values in order to estimate both the particle size and the strain. A comparison between the crystallite size (calculated by both methods) and the strain measured by the WH method is presented in Figure 1C and Table 1. As we can see, the sizes calculated by the two methods do not perfectly align due to the consideration of the strain effect by the WH method, which, therefore, usually yields larger sizes compared to the Scherrer method [47,48]. For the samples with 2 and 10 cycles, the values obtained from WH analysis are only slightly higher with a difference of approximately 15% between the methods; however, in the case of 8 cycles, the gap is approximately 43%. This can be attributed to crystal lattice strain; for nanocrystals formed after two cycles of milling, the strain is very low, as there is a relatively low concentration of such particles, and the majority of the work in the milling process transforms ε-Zn(OH)_2_ into ZnO; however, after additional milling cycles, the number of ZnO particles also increases, and the stress that is placed upon them increases as well, leading to much higher strain (Figure 1C). For example, the strain increases by almost 43% in the transition between 5 and 8 cycles. Another phenomenon—a slight right shift in the diffraction peak position—was also noticed; for example, in the (1,0,1) plane, the peak position shifted from 2θ = 36.05° at 2 cycles to 2θ = 36.1° at 5 cycles (the shift is highlighted in Figure 1B). A right shift in the diffraction peak position indicates a uniform compression strain of the crystal atomic planes [45]. In the 10-cycle sample, the peak position returns to 2θ = 36.05°, implying an energy release between 8 and 10 cycles; therefore, the strain measured after 10 cycles is approximately 8% lower compared with 8 cycles. Such a phenomenon of reduction in nanoparticle strain in ball milling is uncommon, however, it has been demonstrated in long milling processes in ambient conditions [49]. The reduction of strain in such conditions derives from reaching a critical value of dislocation density of the material, resulting in strain relaxation.

Dynamic light scattering (DLS) (Appendix A) and Tauc plots (Appendix A) for the crystals were also measured, and the DLS results showed that the narrowest size distribution is obtained after 10 cycles in which the sample is composed of fairly uniform, ≈500 nm sized, aggregates. The Tauc plots of the powders were used in order to examine the band gap of the ZnO nanopowders, and it was found that almost all of the powders had the same band gap, ≅3.22 eV. This means that the minor change in particle size in our system did not drastically change the electronic properties.

#### 3.1.2. Microscopy 

Electron microscopy of the milled powders was taken by both transmission electron microscopy (TEM) and high-resolution transmission electron microscopy (HRTEM). HRTEM images after 10 cycles are presented in Figure 2; Figure 2A shows a large-scale image of the nanocrystals, revealing that they are small spherical nanoparticles, which are stabilized in the form of large aggregates. It is well known that nanoparticles produced by ball milling are stabilized as aggregates, as no surfactant is added to the synthesis to help to stabilize the particles. Figure 2B presents a close-up image of the ZnO nanocrystal aggregate, clearly showing the lattices of the ZnO crystals; a selected area electron diffraction (SAED) of the nanocrystalline aggregate is inset in the upper right, clearly demonstrating a diffraction pattern fitting a polycrystalline material. An enlarged image of one crystallite that shows a d-spacing of 2.81 angstrom fitting the (1,0,0) plane of ZnO is inset in the bottom right. Images of samples after 2, 5, and 8 cycles were collected by regular TEM microscopy, and the results are presented in Appendix A, respectively.

### 3.2. Phase Transition Mechanism 

#### 3.2.1. TGA Analysis

The mechanism for the mechanochemical transformation of ε-Zn(OH)_2_ into ZnO was investigated through thermal gravimetric analysis (TGA) and X-ray photoelectron spectroscopy (XPS). From the XRD and TEM results, it was already clear that the milling produced ZnO nanoparticles that were stabilized as large aggregates; however, the composition of the powders could not be determined completely by XRD alone, as there could also be amorphous material in the powders that does not diffract. For this reason, we investigated the decomposition of the powders; the TGA experiment comprised three steps: (1) fast heating to 100 °C, (2) an isotherm at 100 °C to remove adsorbed water molecules, and (3) slow heating from 100 to 880 °C. 

The results of the TGA experiment are presented in Figure 3. We identify three major steps in the decomposition: the first at 100 °C is attributed to evaporation of adsorbed water molecules [50], the second starting at 135 °C until around 260 °C is attributed to the decomposition of Zn(OH)_2_ [51], and the third gradually continues up to 880 °C. We believe that this step is attributed to the desorption of ethanol molecules that are adsorbed in the milling process. Vous et al. showed that the desorption of ethanol and 1-propanol adsorbed onto ZnO crystals takes place at high temperatures, due to their ability to adsorb into the crystal lattice [52]. From all of the data presented above, the most significant information that can be extracted is the weight percentage of Zn(OH)_2_ in the powders. After 5 and 8 cycles, there is a non-negligible amount of Zn(OH)_2_, even though the XRD results do not agree; therefore, we conclude that those samples contain amorphous Zn(OH)_2_ (a-Zn(OH)_2_). After 10 cycles, there are only two decomposition steps—water evaporation at 100 °C and a very small weight loss of 6.4%, that starts around 170 °C and ends at 880 °C, which fits the evaporation of ethanol molecules inserted in the ZnO lattice [52]. After 2 cycles, ε-Zn(OH)_2_ was identified by XRD; however, there could also be a small amount of a-Zn(OH)_2_, so we regard the Zn(OH)_2_ phase as a mixture of both crystalline and amorphous material. 

According to TGA analysis, the Zn(OH)_2_ weight percentage is slightly higher after 5 cycles compared with 2 cycles (see Table 2). This could have something to do with the fact that after 2 cycles, both crystalline and amorphous Zn(OH)_2_ are present, which decompose in different ways, and not much is known about the decomposition of a-Zn(OH)_2_ into ZnO. After 8 and 10 cycles, the quantity of a-Zn(OH)_2_ decreases, as more of the material is transformed into uniform ZnO nanocrystals.

#### 3.2.2. XPS Analysis

The mechanism of the transformation was also investigated by XPS analysis. In all cases, the 0 1s XPS spectra of the milled powders presented in Figure 4 show two peaks, centered at 532.3 and 530.6 eV (Figure 4); for ε-Zn(OH)_2_ and after 2, 5, and 8 cycles, only one peak was detected on the surface of the powders at 532.3 eV, which fits the oxygen Zn–OH binding energy in Zn(OH)_2_ [25]. Only after 10 cycles, the second peak at 530.6 eV that fits oxygen binding to Zn in ZnO [25,53] was detected; its presence in only the last sample indicates that the concentration of ZnO in this sample is the highest and is sufficiently high to accumulate on the nanoparticle aggregate surface, thereby providing a peak in the measurement. The presence of the 532.3 eV binding energy in all milled powders indicates the presence of a certain amount of a-Zn(OH)_2_ on the surface of all samples, which is not surprising, as many other researchers have proven that even the purest ZnO nanoparticle powders always contain a hydroxide layer coating the surface of the particles [54,55,56]. The 532.3 eV peak could also be attributed to water molecules that are adsorbed on the surface, as the O–H_2_ binding energy is 532.8 eV [57,58]. The chemical structure of the powders was also investigated by Fourier transform infrared spectroscopy (FTIR), and the results are presented in Appendix A.

Finally, according to the analysis, the mechanochemical mechanism for the transformation can be described; a transition from crystalline to a-Zn(OH)_2_ is followed by a transition from a-Zn(OH)_2_ to crystalline ZnO (zincite):ε-Zn(OH)_2_ (s) ⟶ a-Zn(OH)_2_ (s)
a-Zn(OH)_2_ (s) ⟶ ZnO (s) + H_2_O

### 3.3. Photocatalytic Activity 

So far, we have investigated the size, strain, morphology, phase, aggregation state, and composition of the ZnO milled powders; all of these qualities have an effect on the ZnO activity. As stated above, one of the most common applications of ZnO nanoparticles is photocatalysis; therefore, we have chosen to perform a photodecolorization experiment to the methyl orange (MO) dye using all of the milled powders. MO is an organic pollutant and a potential carcinogen [59] which makes up about 50% of the global dye production [60]. Therefore, research on the photodecolorization and degradation of MO is very important. There are many examples of ZnO nanomaterials being used for the decolorization of MO, and the chemical and physical mechanisms for the catalysis are well known [61,62,63,64,65]; Based on that information, we have chosen the photodecolorization experiment as a model system to test the activity of ZnO. Figure 5A presents the UV-Vis spectra of MO; the purple and yellow lines correspond to spectra before and after exposure to UV light, showing only slight decolorization of the dye (3.3%), while the red, green, blue, and orange lines correspond to MO absorbance after the reaction with the catalyst and exposure to UV light. After 2, 5, 8, and 10 cycles, the extent of MO decolorization was 6.5%, 13.3%, 24.6%, and 40.5%, respectively (Figure 5B); the MO dye concentrations were calculated using the Beer–Lambert law and are presented in Appendix A. The decolorization efficiency increases with the number of cycles, implying a higher concentration of ZnO in the sample. Apart from the catalyst concentration, as stated above, the catalytic activity highly depends on the physical properties of the material, for example, its size, morphology, surface area, and aggregation state of particles. In our case, band gap measurements showed that the slight change in particle size does not have a measurable effect on the electronic properties of the semiconductor (Appendix A); however, the aggregation state of the particles varies greatly from sample to sample.

Initially, after 2 cycles, the concentration of ZnO nanoparticles is relatively low; therefore, the catalytic activity is also low, as expected. After 5, 8, and 10 cycles, the concentration of ZnO increases, and as expected, so does the activity; however, we believe that this increase is also related to the aggregation state of the particles. The catalytic reaction of MO was shown to be much more efficient when the dye is adsorbed upon the surface of the catalyst [66,67], meaning that a larger surface area would lead to higher efficiency. DLS analysis indicates that as the number of cycles increases, the aggregates become more uniform with a small average size of 490 nm; as the surface area of these aggregates is higher compared to larger ones, they are expected to be more efficient catalysts. 

## 4. Discussion

In conclusion, a detailed investigation is presented for the mechanochemical synthesis of ZnO nanoparticles from ε-Zn(OH)_2_ using the high-energy ball milling technique. Traditionally, ZnO nanostructures synthesized through the decomposition of ε-Zn(OH)_2_ were all made in a wet synthetic method by solvating ε-Zn(OH)_2_ crystals in alkaline media; however, quite recently, a dry synthetic route for preparing ZnO crystals from ε-Zn(OH)_2_ was suggested, making the study of dry synthesis important. High-energy ball milling is an efficient top-down synthetic technique for synthesizing nanoparticles, and there are few examples of synthesizing ZnO nanoparticles in a mechanochemical approach; however, the synthesis of ZnO from ε-Zn(OH)_2_ in this approach had never been investigated before. The phase transformation was investigated step by step, as the number of milling cycles was gradually increased, and changes in the composition of the powders as well as in their mechanical and optical properties were examined. While the Zn(OH)_2_ crystallite phase disappeared rapidly in the first milling cycles, a large presence of amorphous Zn(OH)_2_ remained, which decreased only after 10 cycles, indicating that the transformation of ε-Zn(OH)_2_ to ZnO in the dry synthetic route of ball milling goes through amorphization of Zn(OH)_2_, followed by decomposition of the amorphous phase. Overall, our research joins various other reports, which show that the use of high-energy ball milling can successfully produce fine, uniform dispersions of oxide nanoparticles. In the main innovation part of our study, nano-ZnO was successfully prepared from ε-Zn(OH)_2_ crystals, which can be easily synthesized; the structural and optical features of the ZnO nanoparticles may be useful for various applications.

## Figures and Tables

**Figure 1 nanomaterials-11-00238-f001:**
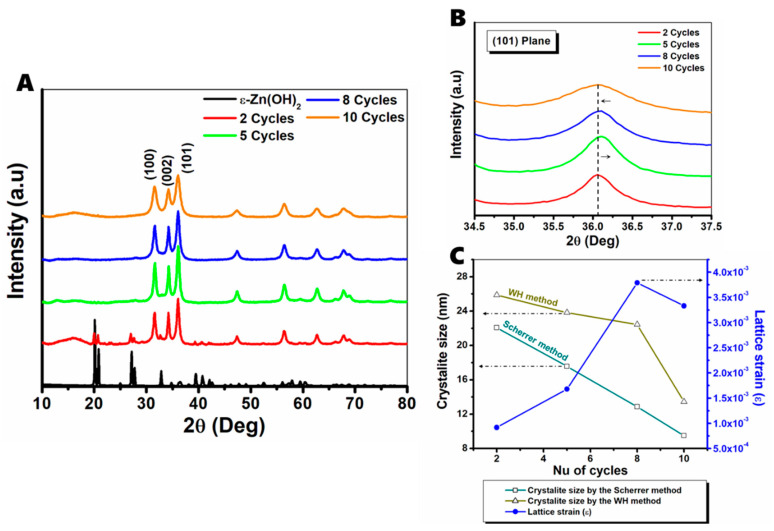
Powder diffraction patterns of zinc oxides: (**A**) XRD spectra of ε-Zn(OH)_2_ (before milling) and milled products with different cycles; (1,0,0), (0,0,2), and (1,0,1) characteristic ZnO peaks are indicated. (**B**) Peak broadening and shifting of the (1,0,1) characteristic peak of ZnO. (**C**) Comparison of crystallite size calculated by the Scherrer and Williamson–Hall (WH) methods and lattice strain.

**Figure 2 nanomaterials-11-00238-f002:**
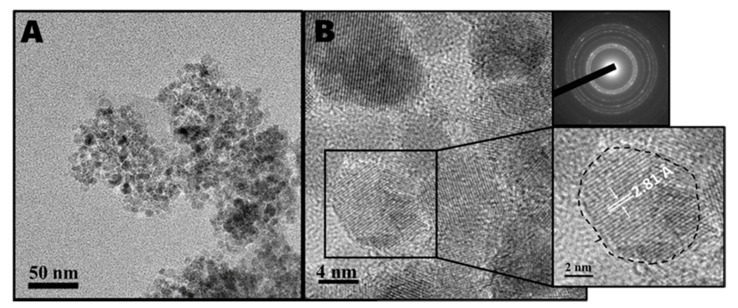
HRTEM images after 10 cycles indicating ZnO nanoparticle aggregates: (**A**) Large-scale image showing an aggregate consisting of small ZnO crystallites. (**B**) Small-scale image of nanocrystallites highlighting a crystallite with d-spacing of 2.81 angstrom fitting the (1,0,0) plane of ZnO (bottom right inset), and a SAED pattern of nanocrystalline aggregates (upper right inset).

**Figure 3 nanomaterials-11-00238-f003:**
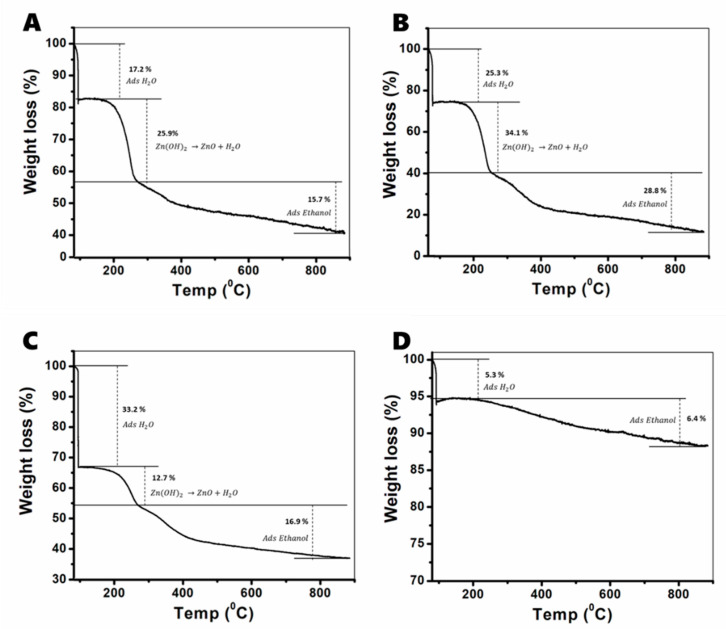
Thermal gravimetric analysis of ZnO milled powders—a detailed analysis of the decomposition steps of samples after 2 (**A**), 5 (**B**), 8 (**C**), and 10 (**D**) cycles.

**Figure 4 nanomaterials-11-00238-f004:**
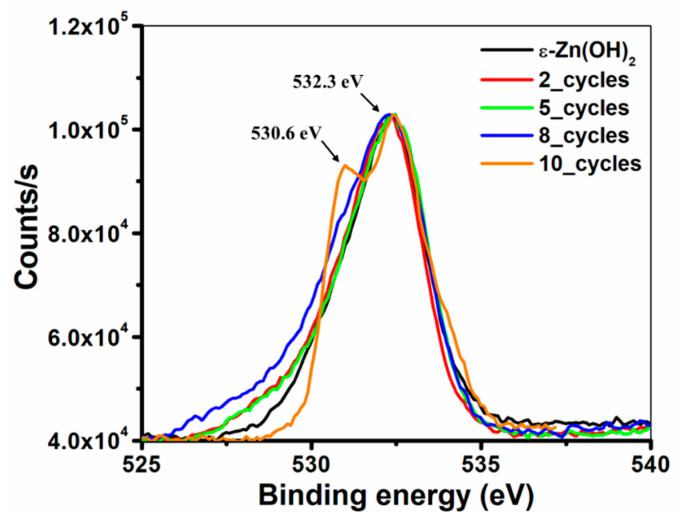
X-ray photoelectron spectroscopy (XPS) analysis of ZnO milled powders—0 1s XPS spectra of ε-Zn(OH)_2_ and milled powders examining the oxygen–zinc binding energy.

**Figure 5 nanomaterials-11-00238-f005:**
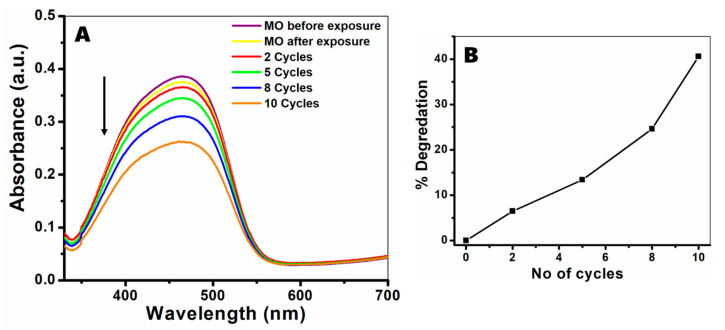
Methyl orange (MO) degradation assay. (**A**) Absorbance spectra of MO before and after exposure to UV radiation for 30 min (purple and yellow lines) and in the presence of powders after 2 (red), 5 (green), 8 (blue), and 10 (orange) cycles. (**B**) Degradation percentage of MO compared to the dye before UV exposure. Dye concentrations were calculated using the Beer–Lambert law.

**Table 1 nanomaterials-11-00238-t001:** ZnO crystallite sizes calculated by the Scherrer and Williamson–Hall (WH) methods, and strain values calculated using the WH method.

Cycles	Average FWHM (deg)	Crystallite Size (nm)		Strain (ε)
		Scherrer	Williamson-Hall	
**2**	0.54896	22.07	25.87	9.17$\;\times 10^{-4}$
**5**	0.62866	17.57	23.82	1.68$\;\times 10^{-3}$
**8**	0.87176	12.86	22.44	3.79$\;\times 10^{-3}$
**10**	1.14171	9.51	13.45	3.33$\;\times 10^{-3}$

**Table 2 nanomaterials-11-00238-t002:** Weight percentage of Zn(OH)_2_ in ZnO milled powders calculated from TGA.

No of Cycles	Phase	Zn(OH)_2_ Percentage
**2**	ε-Zn(OH)_2_ + a-Zn(OH)_2_	25.9
**5**	a-Zn(OH)_2_	34.1
**8**	a-Zn(OH)_2_	12.7
**10**	a-Zn(OH)_2_	0

## Data Availability

Data is contained within the article or Appendix A. Further, the published data can be reused by appropriate acknowledgement.

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
