# Peer review of "Solvent-Free Mechanochemical Synthesis of ZnO Nanoparticles by High-Energy Ball Milling of ε-Zn(OH)2 Crystals"

_nanomaterials, 2021, doi:10.3390/nano11010238_

Round 1
Reviewer 1 Report
In this work a detailed investigation of solvent-free mechanochemical synthesis of zinc oxide nanoparticles from ε-Zn(OH)2 crystals by high energy ball milling in ambient condition is presented.
It is indeed an interesting approach for synthesizing ZnO from ε-Zn(OH)2, which, to the best of my knowledge this approach has never been investigated before; it offers a simple and cheap solvent-free way to produce ZnO nanoparticles at a ambient temperature.
The ZnO nanoparticles have been studied regarding their size and strain, optical properties, aggregate size, etc., using XRD, TEM, DLS, UV-Vis and FTIR.
The authors should discuss more some issues.
-Why the lattice strain decreases with 10 cycles ?
The strain increases up to 8 cycles. How do the authors explain the change for 10 cycles? Please discuss further this behavior.
-Regarding the photocatalysis, the authors actually do not see the degradation of MO, as they state. What they observe is its decolorization. Please rephrase and use this term instead.
-The authors present the absorption spectra of MO upon UV irradiation for 30min. The UV source they use is centered at 254nm. Which is the intensity of the UV source in W/cm2?
-The authors should should discuss the kinetics of MO decolorization. They should present the absorption of MO for at least one sample for several time intervals and calculate the apparent value of K (for example, =-lnC/Co, for 1st order kinetics) .
-Moreover, the authors should check the use of their samples in terms of their photocatalytic activity. They should use the more efficient sample at least for 3 to 5 times, and calculate the k value. We can compare the effectiveness of the samples only by using the kinetics.
This work should be accepted for publication after working with the above issues.
Author Response
Comments of reviewer #1
“In this work a detailed investigation of solvent-free mechanochemical synthesis of zinc oxide nanoparticles from ε-Zn(OH)2 crystals by high energy ball milling in ambient condition is presented. It is indeed an interesting approach for synthesizing ZnO from ε-Zn(OH)2, which, to the best of my knowledge this approach has never been investigated before; it offers a simple and cheap solvent-free way to produce ZnO nanoparticles at a ambient temperature. The ZnO nanoparticles have been studied regarding their size and strain, optical properties, aggregate size, etc., using XRD, TEM, DLS, UV-Vis and FTIR.”
The authors should discuss more some issues. Why the lattice strain decreases with 10 cycles ?"
Response
We thank the reviewer for this comment in view of this comment in the revised article we have added a detailed discussion with our explanation lattice strain decreases (please see section 3.1.1 lines: 182-192)
Comments of reviewer #1
“The strain increases up to 8 cycles. How do the authors explain the change for 10 cycles? Please discuss further this behavior.”
Response
As mentioned above, in section 3.1.1 of the revised article a detailed explanation of the lattice strain change as a function of particles size is given.
Comments of reviewer #1
“Regarding the photocatalysis, the authors actually do not see the degradation of MO, as they state. What they observe is its decolorization. Please rephrase and use this term instead. The authors present the absorption spectra of MO upon UV irradiation for 30min. The UV source they use is centered at 254nm. Which is the intensity of the UV source in W/cm2?”
Response
We thank the reviewer the term degradation has been replaced with decolorization, and details on the intensity of the UV source were added to the article.
Comments of reviewer #1
“The authors should should discuss the kinetics of MO decolorization. They should present the absorption of MO for at least one sample for several time intervals and calculate the apparent value of K (for example, =-lnC/Co, for 1st order kinetics) .Moreover, the authors should check the use of their samples in terms of their photocatalytic activity. They should use the more efficient sample at least for 3 to 5 times, and calculate the k value. We can compare the effectiveness of the samples only by using the kinetics.”
Response
The reviewer makes a correct claim that we agree with regarding the kinetic measurements of the MO decolorization. However, in this article, we focus on the new synthesis of ZnO nanoparticles by high energy ball milling of ε-Zn(OH)2 while the photoactive measurements of MO decolorization with ZnO nanoparticles are just an example of the uses of the particles and not the main issue in this article. The measurements that the reviewer requests are important but require a great deal of additional research work and are time-consuming. Since this topic of the kinetics of MO decolorization by ZnO nanoparticles of the not the main point we, therefore, assume that performing the additional measurements as asked by the reviewer are essential or necessary for understanding the main issue of the article. In the future, we will continue our research on this topic and will be happy to present more details on the kinetics of MO decolorization by ZnO nanoparticles.

Reviewer 2 Report
I have reviewed the paper.The result is interesting. The minor revision is needed and these are my comments.
1) Authors should improve the English quality sufficient for publication.
2) The abstract and conclusion should be rewritten to reflect the important findings in the work.
3) Please compare your work with similar work and what is the good point in your work.
Author Response
Comments of reviewer #2
“I have reviewed the paper. The result is interesting. The minor revision is needed and these are my comments. 1) Authors should improve the English quality sufficient for publication.”
Response
We have carefully read the article again, all spelling errors have been corrected and the article has been professionally edited in English.
Comments of reviewer #2
“2) The abstract and conclusion should be rewritten to reflect the important findings in the work.”
Response
We have written the abstract and conclusion parts of the article to reflect the importance of our findings as aksed by the reviewer.
Comments of reviewer #2
“3) Please compare your work with similar work and what is the good point in your work”
Response
We thank the reviewer for this comment, in the revised article we compare your work with other reported papers in the field of our study.

Reviewer 3 Report
This paper describes mechanochemical preparation of ZnO from Zn(OH)2.
Basically, there were no serious problems in the description, it should be
accepted almost as it is.
However, explanation of Section 3.2 (Lines 292-307) was difficult to
understand, please re-write not to be like "excuse".
That's all.
Author Response
Comments of reviewer #3
“This paper describes mechanochemical preparation of ZnO from Zn(OH)2. Basically, there were no serious problems in the description, it should be accepted almost as it is. However, explanation of Section 3.2 (Lines 292-307) was difficult to understand, please re-write not to be like "excuse".”
Response
We thank the reviewer and we re-write Section 3.2 as asked by the reviewer.

Round 2
Reviewer 1 Report
The authors have commented on all the revisions. They have worked anf fixed most of the issues.
The authors have commented on all the revisions. They have worked anf fixed most of the issues.
This manuscript can be published.